# Wearing WHOOP More Frequently Is Associated with Better Biometrics and Healthier Sleep and Activity Patterns

**DOI:** 10.3390/s25082437

**Published:** 2025-04-12

**Authors:** Gregory J. Grosicki, Finnbarr Fielding, Jeongeun Kim, Christopher J. Chapman, Maria Olaru, William von Hippel, Kristen E. Holmes

**Affiliations:** 1Performance Science, WHOOP Inc., Boston, MA 02215, USA; greg.grosicki@whoop.com (G.J.G.); finn.fielding@whoop.com (F.F.); janekim319@gmail.com (J.K.); christopher.chapman@whoop.com (C.J.C.); bill.vonhippel.contractor@whoop.com (W.v.H.); 2Research Algorithms and Development, WHOOP Inc., Boston, MA 02215, USA; maria.olaru@whoop.com; 3Research with Impact, Brisbane, QLD 4000, Australia

**Keywords:** wearable technology, resting heart rate, heart rate variability, smart phone, behavior change

## Abstract

Wearable devices are increasingly used for health monitoring, yet the impact of consistent wear on physiological and behavioral outcomes is unclear. Leveraging nearly a million days and nights of longitudinal data from 11,914 subscribers, we examined the associations between the frequency of wearing a wrist-worn wearable device (WHOOP Inc., Boston, MA, USA) and 12-week changes in biometric, sleep, and activity profiles, modeling both between- and within-person effects. Higher average wear frequency and week-to-week increases in wear were associated with a lower resting heart rate (RHR), higher heart rate variability (HRV), longer and more consistent sleep, and greater weekly and daily physical activity duration (Ps < 0.01). A within-person multiple mediation analysis indicated that increased sleep duration partially mediated the association between wear frequency and a standardized (z-scored) RHR (indirect effect = −0.0387 [95% CI: −0.0464, −0.0326]), whereas physical activity minutes did not (indirect effect = 0.0003 [95% CI: −0.0036, 0.0040]). A Granger causality analysis revealed a modest but notable association between prior wear frequency and future RHR in participants averaging ≤5 days of weekly wear (*p* < 0.05 in 10.92% of tests). While further research is needed, our findings provide real-world evidence that sustained wearable engagement may support healthier habits and improved physiological outcomes over time.

## 1. Introduction

Wearable technology use has grown exponentially, with nearly one in three Americans using a wearable device [1], such as the WHOOP strap (WHOOP Inc., Boston, MA, USA), to monitor key physiological markers like the resting heart rate (RHR) and heart rate variability (HRV), as well as health behaviors like sleep and physical activity [2]. While these devices are designed for continuous wear, adherence varies widely [1,3]. Despite assumptions that greater wear frequency enhances self-monitoring and improves health outcomes, research on the impact of wear consistency over time remains limited [4,5,6], with most studies focusing on individuals with underlying health conditions [7].

Consistent engagement with wearable devices may reinforce positive health habits and improve physiological markers. Indeed, wearable monitoring has been linked to increased physical activity [6,8], reduced sedentary time [6], better sleep quality [9,10], and improved cardiovascular health [8]. More frequent wear may enhance data continuity, providing a more complete and timelier picture of health metrics, which in turn strengthens the feedback loop between monitoring, behavior adjustments, and health improvements. To this end, WHOOP uniquely leverages artificial intelligence to deliver personalized, actionable insights, helping users to optimize recovery, training, and sleep behaviors based on their unique individual physiological responses. However, the extent to which WHOOP wear frequency influences changes in biometrics, sleep, and activity remains unclear.

This real-world evidence study evaluated associations between WHOOP wear frequency and 12-week changes in biometric and behavioral profiles using a large longitudinal dataset encompassing nearly a million days and nights of data. We hypothesized that wearing WHOOP more consistently would be associated with a lower RHR and higher HRV, longer and more consistent sleep, and increased physical activity levels, thereby reinforcing the role of wear frequency in maximizing the benefits of wearable technology.

## 2. Materials and Methods

### 2.1. Participant Eligibility

All participants consented to the use of their anonymized data for research purposes. We analyzed biometric, sleep, and physical activity data from 12,000 randomly selected individuals (6000 males and 6000 females) who purchased a WHOOP subscription (WHOOP Strap 4.0, WHOOP Inc., Boston, MA, USA) between 1 January 2024 and 15 November 2024. Data were evaluated for 12 weeks (84 days) for each member, with eligibility requiring at least one complete cycle day (≥85% of data from primary sleep episode) in both Week 1 and Week 12 post-activation. To account for seasonal trends in biometric and behavioral characteristics [11], members were evenly distributed throughout the year (~1000 every two months per sex).

### 2.2. Data Collection

The WHOOP strap continuously records heart rate via photoplethysmography (PPG) and movement via a three-axis accelerometer. Key cardiovascular metrics, RHR and HRV, were extracted as a weighted average from the primary sleep episode [12]. HRV, more precisely pulse rate variability, due to its derivation from PPG signals [13], was calculated using the root mean square of successive differences (RMSSD) [14]. Physical activity metrics included the amount of time spent in four physical activity zones, classified as a percentage of age-predicted heart rate maximum [15]: zone 2 (60–70%), zone 3 (70–80%), zone 4 (80–90%), and zone 5 (90–100%). Daily physical activity was represented as the sum of the total time spent in each activity zone, with higher-intensity activities (zones 4 and 5) being weighted by a factor of two [16]. Physical activity variables of interest included cumulative total weekly physical activity duration, as well as the average duration of physical activity per wear day, to normalize for differences in adherence. Sleep metrics included total sleep duration and sleep consistency, characterized as regularity of sleep onset and offset times over a 4-day window, with greater weighting for more recent days [17]. WHOOP cardiovascular and sleep measures were validated against gold-standard electrocardiogram and polysomnography, demonstrating a low degree of bias and high accuracy (e.g., <20 min bias and precision errors for sleep duration; 0.7 beats per minute for heart rate; 4.7 ms bias for heart rate variability) [9,18,19]. Because data were anonymized and securely stored, this study was deemed exempt from Institutional Review Board (IRB) oversight by Salus’s IRB (#6483).

### 2.3. Categorizing Participants by Wear Frequency

After performing data cleaning to remove incomplete or invalid entries, the final dataset comprised 907,249 days/nights of data from 11,914 individuals. Participants were categorized into four wear frequency groups based on their average weekly wear throughout the 12-week study period (see Table 1). More frequent wear was associated with a lower weekday percentage, a higher proportion of male participants, an older age, and lower BMI (Ps < 0.05). Additionally, participants who wore WHOOP more frequently had superior baseline (i.e., Week 1) biometric, sleep, and activity characteristics compared to those with less frequent wear. Participants who wore WHOOP daily had a 3.769 bpm lower RHR [95% CI: −3.997, −3.54], 0.615 h longer [95% CI: 0.571, 0.658] and 11.416 percentage points of more consistent sleep [95% CI: 10.258 to 12.574], and accumulated more weekly (89.750 min [95% CI: 72.741, 106.759]) and daily activity (10.149 min [95% CI: 8.989, 11.308]) compared to those wearing WHOOP < 5 days per week.

To complement between-person groupings, week-to-week deviations in individual wear frequency were captured using person-mean centering and classified into five categories: (i) “Much Less Than Usual”, indicating wear frequency 2 days fewer than typical (≤−2.0 days; n = 5268 weeks); (ii) “Slightly Less Than Usual”, indicating wear frequency between 2.0 and 0.1 days fewer than typical (−2.0 < deviation < −0.1 days; n = 22,837 weeks); (iii) “Typical”, indicating wear frequency within ±0.1 of the typical value (−0.1 ≤ deviation ≤ 0.1 days; n = 54,253 weeks); (iv) “Slightly More Than Usual”, indicating wear frequency between 0.101 and 1.0 days more than typical (0.101 ≤ deviation ≤ 1.0 days; n = 49,290 weeks); and (v) “Much More Than Usual”, indicating wear frequency 1.0 days more than typical (>1.0 days; n = 7992 weeks). Thresholds were determined to ensure that there was a sufficient proportion (~5%) of weeks in each category, with different cut-off points used for the “Much Less” (≤−2.0 days) and “Much More” (>1.0 days) categories due to the limited number of weeks with wear frequency of 2 or more days greater than typical (n = 1154).

### 2.4. Statistical Analysis

All statistical analyses were conducted using Python (version 3.11.9). Participant descriptive characteristics are presented as means ± standard deviation (SD), and normality was assessed using the Shapiro–Wilk test and a visual inspection of quantile–quantile plots.

Descriptive characteristics, biometrics, sleep, and activity variables from the initial week of the study (i.e., baseline) were compared among groups using analysis of variance (ANOVA) with Tukey’s post hoc test to explore specific group differences.

To examine associations between WHOOP wear with biometric, sleep, and activity outcomes over the 12-week study period, we modeled both between-person (each participant’s average weekly wear frequency) and within-person effects (week-to-week deviations from their individual average, captured via person-mean centering). Models were estimated using both continuous and categorical representations (see Section 2.3 in Materials and Methods) of these effects, with primary analyses focusing on continuous models and categorical models employed for visualization. All models were adjusted for the baseline value of the outcome, as well as age, sex, BMI, season, and weekday percentage. Random intercepts accounted for within-person correlations, while random slopes for time allowed for individual variability in outcome trajectories over the 12-week period.

To investigate mechanisms underlying associations between wear frequency and RHR, we conducted a within-person multiple mediation analysis using linear mixed-effects models. The analysis assessed whether week-to-week changes in total physical activity and average sleep duration contributed to this association. Bootstrapped confidence intervals were used to assess the statistical significance of indirect effects.

To evaluate temporal relations between WHOOP wear frequency and biometric changes, we conducted a Granger causal analysis to determine whether past wear frequency predicted future RHR. The analysis focused on participants who wore WHOOP five days per week or less to maximize variability in wear frequency. Since recent wear (i.e., days vs. weeks) was hypothesized to be particularly relevant for predicting future RHR, wear time was computed as a 7-day rolling average. We then applied Granger causality testing with a maximum lag of five days using the rolling average of daily wear to predict z-scored RHR. Statistical significance was set at α ≤ 0.05 for all analyses.

## 3. Results

### 3.1. Higher Wear Frequency and Week-to-Week Increases in Wear Associated with Better Biometrics

The results of the models estimating RHR and HRV using continuous between- and within-person effects are provided in Table 2. The RHR increased modestly over time (*p* < 0.001), but more consistent WHOOP wear mitigated this effect (Ps ≤ 0.013; Figure 1). Additionally, higher average wear frequency and week-to-week increases in wear were associated with a lower RHR and higher HRV (Ps ≤ 0.002).

Categorical models showed stepwise benefits of WHOOP wear on the RHR (Figure 2A,B). The RHR was estimated to be 0.933 bpm lower [95% CI: −1.191, −0.675] in those who wore WHOOP every day compared to those who wore it <5 days per week (Figure 2B). Similarly, weeks classified as “Much More Than Usual” were associated with an estimated −1.931 bpm reduction [95% CI: −2.064, −1.798] compared to “Much Less Than Usual” (Figure 2A).

HRV was comparable across weekly wear categories (Ps ≥ 0.119; Figure 2D). However, HRV was estimated to be 1.214 ms higher [95% CI: 0.904, 1.524] in weeks classified as “Much More Than Usual” compared to “Much Less Than Usual” (Figure 2C).

### 3.2. Sleep Consistency Improves over Time, and Higher Wear Frequency and Week-to-Week Increases in Wear Are Associated with Longer and More Consistent Sleep

The results from the models of sleep duration and consistency are provided in Table 3. Sleep consistency improved over time (*p* = 0.007), while duration remained stable (*p* = 0.956). A higher average weekly wear frequency and week-to-week increases in wear were associated with longer and more consistent sleep (Ps < 0.001).

The categorical models showed stepwise benefits of WHOOP wear on sleep variables (Figure 3). Compared to participants who wore WHOOP < 5 days per week, those who wore it every day had longer sleep duration (+0.334 hrs [95% CI: 0.283, 0.385]; Figure 3B) and greater sleep consistency (+4.596 percentage points [95% CI: 3.784, 5.408]; Figure 3D). Similarly, weeks classified as “Much More Than Usual” regarding WHOOP wear were associated with longer sleep duration (+0.253 h [95% CI: 0.223, 0.282]; Figure 3A) and greater sleep consistency (+3.504 percentage points [95% CI: 3.089, 3.918]; Figure 3C) relative to weeks classified as “Much Less Than Usual”.

### 3.3. Physical Activity Increases over Time, and Higher Wear Frequency and Week-to-Week Increases in Wear Are Associated with More Activity

The results from the models of total weekly and daily activity minutes (i.e., weekly activity minutes indexed to the number of active days per week) are provided in Table 4. The weekly average physical activity and daily activity minutes both increased over time (Ps ≤ 0.009). Higher average wear frequency and week-to-week increases in wear time were both associated with increased weekly total and daily average physical activity minutes (Ps < 0.001).

Once again, categorical models showed stepwise benefits of WHOOP wear on weekly total and daily average physical activity minutes (Figure 4). Compared to those who wore WHOOP < 5 days a week, those who wore WHOOP every day were estimated to have 90.948 min greater weekly total physical activity [95% CI: 80.653, 101.244; Figure 4B] and 9.995 more daily activity minutes [95% CI: 7.750, 12.239; Figure 4D]. Likewise, weeks classified as “Much More Than Usual” were associated with 124.099 more min of physical activity [95% CI: 119.240, 128.959; Figure 4A] and 4.400 more min of activity per wear day [95% CI: 3.621, 5.180; Figure 4C] compared to weeks classified as “Much Less Than Usual”.

### 3.4. Sleep Duration Partially Mediates the Association Between Wear Frequency and RHR

To further explore the association between wear frequency and RHR, we conducted a within-person multiple mediation analysis to assess whether week-to-week changes in wear frequency were linked to week-to-week changes in standardized (z-scored) RHR through changes in sleep duration or total physical activity. Sleep duration partially mediated this association (indirect effect = −0.0387 [95% CI: −0.0464, −0.0326]), whereas physical activity minutes did not (indirect effect = 0.0003 [95% CI: −0.0036, 0.0040]). A significant direct effect of week-to-week wear time variation on standardized RHR remained (β = −0.3676, *p* < 0.001).

### 3.5. Past Wear Frequency Predicts Future Resting Heart Rate

The results from the Granger causality tests indicate a modest but notable association between prior wear frequency and future RHR. On average, 10.92% of tests yielded statistically significant results (Figure 5), suggesting that variability in wear frequency has a non-random association with subsequent fluctuations in RHR. The proportion of significant *p*-values over the five-day lag ranged from 10.525% (Day 5) to 11.446% on Day 3. Importantly, higher 7-day rolling averages were associated with lower next-day z-scored RHR values (β = −0.008 [95% CI: −0.010, −0.006]). When the analysis was reversed to test whether past RHR predicted future wear time, the proportion of significant tests was quantitatively lower, ranging from 7.4 to 9.7%.

## 4. Discussion

Leveraging nearly a million days and nights of longitudinal data, we ran between-person comparisons and within-person longitudinal models to comprehensively understand how the frequency of wearing WHOOP relates to members’ health metrics, sleep, and activity patterns during the first 12 weeks of device use. We found that wearing WHOOP more frequently was associated with a lower RHR, higher HRV, and healthier sleep and activity patterns. These findings provide compelling initial evidence that consistent engagement with WHOOP is linked to physiological and behavioral benefits.

Individuals who wore WHOOP more consistently tended to have markedly healthier biometric and behavioral profiles than those with lower wear times. This is particularly noteworthy given that even individuals in the lowest wear time group exhibited impressively low RHR profiles and, on average, met the 150 min weekly physical activity guideline [20]. Nonetheless, the baseline RHR in the every day wear group was nearly 4 bpm lower, a difference of potential clinical significance given evidence linking a 1 bpm increase to a 3% higher risk for all-cause mortality and a 2% higher risk for coronary heart disease [21]. Importantly, even after adjusting for baseline values and other key covariates such as age, sex, and BMI, wear group RHR differences persisted, providing increased confidence in the association between wear frequency and cardiovascular health. One possible explanation for these differences is that individuals inclined to wear a health monitor may be inherently more health-conscious or physically active, leading to superior biometric profiles. Indeed, individuals in the highest wear group were at the upper end of the activity spectrum and had longer sleep durations that approached the optimal range for longevity [22]. However, the findings from our Granger causal analysis show that greater past wear time predicts lower future RHR, suggesting that frequent wear may itself reinforce engagement in health-promoting behaviors. In sum, these findings suggest that sustained wearable engagement is linked not only to baseline health differences but also to positive behavioral reinforcement, highlighting the potential benefits of consistent WHOOP use for cardiovascular and overall health.

Beyond cross-sectional differences between wear groups, our analyses revealed significant within-person benefits of wearing WHOOP, as indicated by superior biometric, sleep, and activity profiles in weeks where WHOOP was worn more frequently. These analyses strengthen the evidence for a genuine benefit of wearing WHOOP since they account for inter-individual characteristics, essentially using each person as their own control. Most comparably, a previous randomized controlled trial showed improvements in self-reported sleep quality after just one week of wearing WHOOP [9]. Although self-reported sleep characteristics were unavailable in the present study, an improvement in sleep consistency was observed. While sleep duration is widely recognized as a key health behavior, sleep consistency is emerging as an equally, if not more, salient predictor of health and well-being than sleep duration [23], with relevance even in young and apparently healthy individuals [24]. Our findings, in concert with previous research [9], demonstrate the efficacy of WHOOP in improving both subjective and objective sleep outcomes. Similarly, a recent clinical study demonstrated that participants fell asleep faster and perceived sleep quality improvements with a noise-masking digital wearable device [10], further reinforcing the role of wearable technology as a tool for optimizing sleep quality. Notably, sleep was recently recognized as an essential component of cardiovascular health with its inclusion in the American Heart Association’s “Life’s Essential 8”, underscoring the value of optimizing sleep as a key determinant of health span [25].

Consistent with prior research showing that activity trackers appear to be effective at increasing physical activity [26], we observed increases in both weekly and daily activity durations over the 12-week study period. Moreover, weeks of higher WHOOP wear were characterized by increased weekly and daily physical activity. Interestingly, however, week-to-week variations in sleep duration, but not activity duration, mediated the association between increased wear frequency and a lower RHR. While regular exercise is well documented to lower the RHR over time [27], acute increases in exercise can temporarily elevate the RHR [28], which may explain the lack of a significant mediation effect for activity duration in our study. Nonetheless, improvements in sleep consistency and activity duration, combined with evidence linking higher weekly wear to increased sleep duration and reduced RHR, provide compelling evidence for the efficacy of WHOOP to reinforce health-promoting behaviors.

A key but unavoidable limitation of this study is the lack of true pre-WHOOP values for biometrics, sleep, and activity. Individuals likely modified their habitual lifestyle and activity habits immediately upon starting WHOOP, meaning that Week 1 biometrics may not fully represent pre-WHOOP values. This phenomenon, often seen in self-monitoring interventions [29], highlights the need for future research to better understand initial behavioral changes upon adopting wearable technology, as well as strategies to sustain early improvements over time in diverse populations of varying ages, sexes, and health statuses. Despite this limitation, our findings still demonstrate clear biometric and behavioral benefits of WHOOP use, which perhaps are all the more impressive in this context.

Our study should be interpreted in the context of both its strengths and limitations. First, as an observational study, we cannot eliminate confounding factors or establish causation despite our use of within-person models and temporal analysis. Individuals who chose to wear WHOOP more frequently may differ in other important ways, such as baseline fitness level (e.g., maximal oxygen consumptions, VO_2_max) or underlying motivation to improve health and fitness, which we were unable to quantify in this study. Additionally, while within-person analyses helped to mitigate between-person confounding factors, it is possible to determine time-varying factors such as an individual wearing their WHOOP more while training for a marathon and related activity and biometric changes not explained by WHOOP itself. Second, because our study relied on wearable-collected data, measures were contingent on device wear. Although unlikely, it is possible that participants with lower wear frequency missed recording “healthier” days, potentially biasing associations between wear time and biometric and behavioral characteristics. Additionally, we were unable to account for the possibility that some individuals removed their WHOOP devices during the day, leading to unrecorded activities. Third, while HRV is commonly used to refer to the metric derived from WHOOP, it is more accurately termed PRV given its derivation from PPG signals rather than electrocardiography. Although PRV approximates HRV under resting and free-living conditions, the two are not identical, particularly when vascular properties fluctuate [30]. This distinction is important when interpreting the findings, especially in dynamic physiological states. Fourth, our sample consisted solely of WHOOP subscribers, meaning participants had the motivation and resources to invest in a wearable technology subscription. This selection bias might limit the generalizability of our findings to broader populations, such as those with lower technological literacy. Finally, our study was limited to 12 weeks, and engagement with the device may decline over time as the novelty wears off. Future research should examine longer-term adherence trajectories, segmenting users by characteristics such as age and sex, and identifying behavioral changes underlying biometric health improvements to inform more targeted and personalized recommendations.

Despite these limitations, our study had many notable strengths. We leveraged a large, real-world dataset with continuous physiological monitoring, enhancing ecological validity. The combination of between- and within-person effects, within-person multiple mediation models, and bi-directional Granger causality analysis testing facilitated a comprehensive evaluation of relations between wear frequency, behavior, and biometric outcomes. Last, this study is among the first to systematically demonstrate a dose–response relation between wearable engagement and improvements in user behavior and health metrics, contributing valuable insights to the growing field of digital health monitoring.

In conclusion, we demonstrate that wearing WHOOP more often is associated with better biometric indices and healthier sleep and activity patterns, both across different individuals and within the same individual over time. These findings suggest that wearable technology, and WHOOP in particular, functions not just as a passive tracking tool but also as an active facilitator of meaningful behavior change. Future randomized controlled trials should explore strategies that increase wearable exposure over time and assess how personalized insights from these devices can be used for preventive and/or therapeutic strategies. Meanwhile, our study provides real-world evidence that sustained use of a wearable device like WHOOP may be a simple yet effective component of a healthier lifestyle, contributing to improved sleep, more activity, and better physiological health profiles.

## Figures and Tables

**Figure 1 sensors-25-02437-f001:**
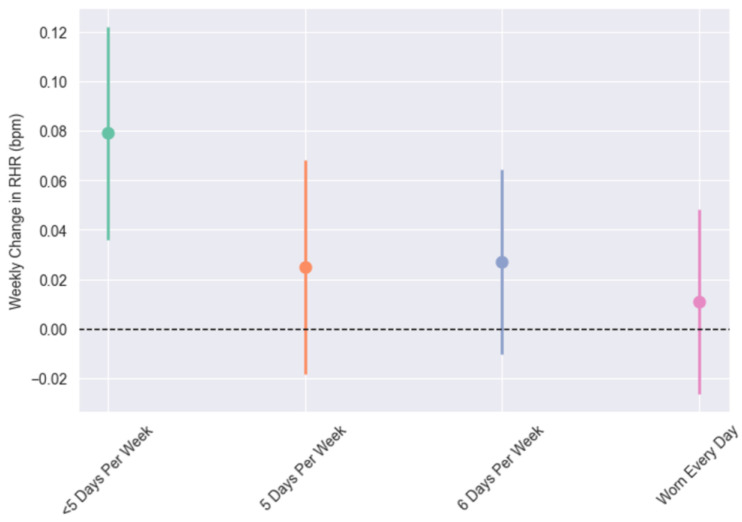
Weekly changes in resting heart rate (RHR) from categorical models across wear groups: “<5 days per week” was defined as weekly wear average of <5 days across 12-week study period, “5 days per week” was defined as 5.0–5.99, “6 days per week” as 6.0–6.99, and “Worn Every Day” as 7 days of wear each week, with one-day exception in first week to account for activation technicalities. All groups were significantly lower than <5 days per week (Ps ≤ 0.013).

**Figure 2 sensors-25-02437-f002:**
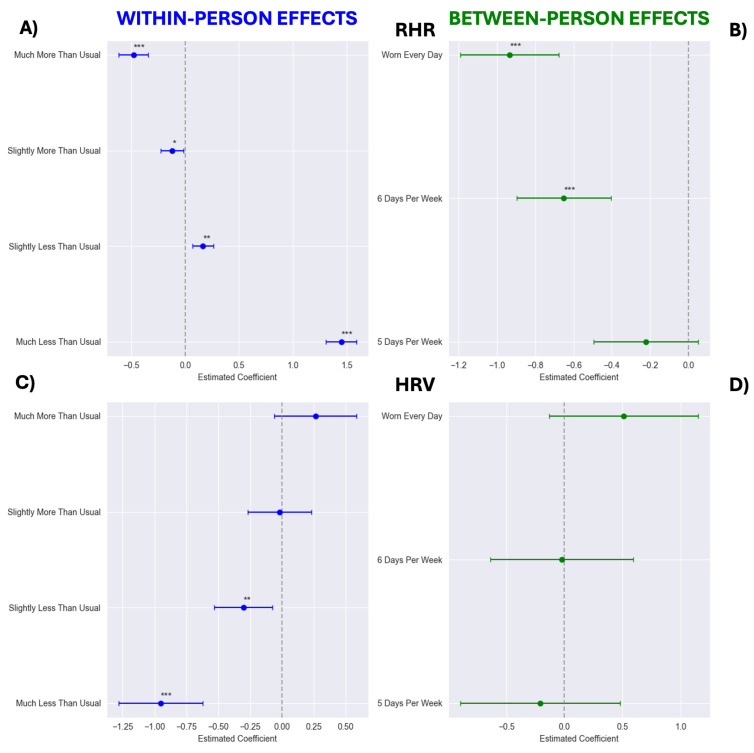
Between- (panels (**B**,**D**)) and within-person (panels (**A**,**C**)) effect estimates from categorical models of wearing WHOOP on resting heart rate (RHR; panels (**A**,**B**)) and heart rate variability (HRV; panels (**C**,**D**)). Within-person reference categories modeled as follows: “Typical Wear”, wear frequency within ±0.1 of typical (−0.1 ≤ deviation ≤ 0.1); “Much Less Than Usual”, wear frequency 2 days fewer than typical (≤−2.0 days); “Slightly Less Than Usual”, wear frequency between 2.0 and 0.1 days fewer than typical (−2.0 < deviation < −0.1 days); “Slightly More Than Usual”, wear frequency between 0.101 and 1.0 days more than typical (0.101 ≤ deviation ≤ 1.0 days); “Much More Than Usual”, wear frequency 1.0 days more than typical (>1.0 days). For between-person models, <5 days per week served as reference. Wear time of “5 days per week” was defined as 5.0–5.99, “6 days per week” as 6.0–6.99, and “Worn Every Day” as 7 days of wear each week, with one-day exception in first week to account for activation technicalities. * *p* < 0.05, ** *p* < 0.01, and *** *p* < 0.001.

**Figure 3 sensors-25-02437-f003:**
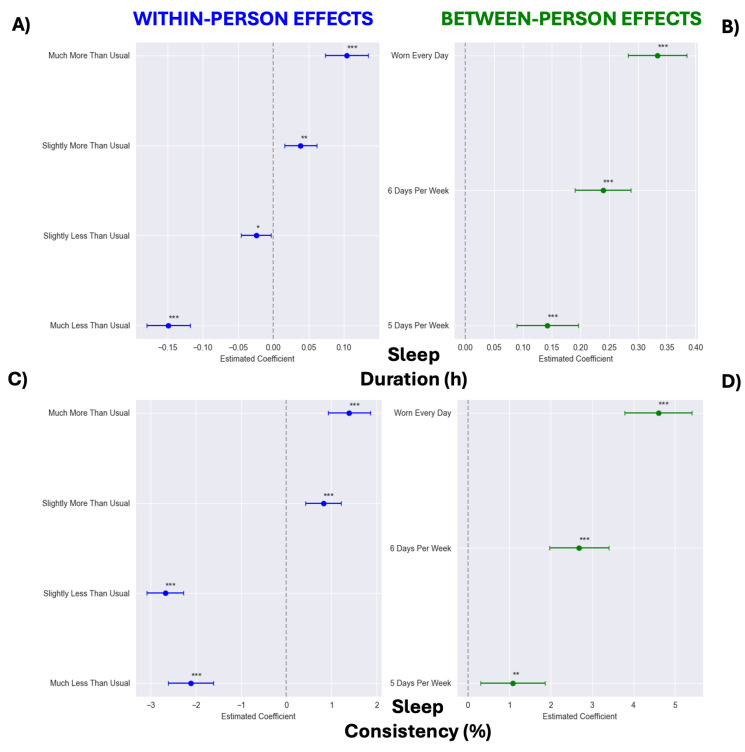
Between- (panels (**B**,**D**)) and within-person (panels (**A**,**C**)) effect estimates from categorical models of wearing WHOOP on sleep duration (panels (**A**,**B**)) and sleep consistency (panels (**C**,**D**)). Within-person reference categories modeled as follows: “Typical Wear”, wear frequency within ±0.1 of typical (−0.1 ≤ deviation ≤ 0.1); “Much Less Than Usual”, wear frequency 2 days fewer than typical (≤−2.0 days); “Slightly Less Than Usual”, wear frequency between 2.0 and 0.1 days fewer than typical (−2.0 < deviation < −0.1 days); “Slightly More Than Usual”, wear frequency between 0.101 and 1.0 days more than typical (0.101 ≤ deviation ≤ 1.0 days); “Much More Than Usual”, wear frequency 1.0 days more than typical (>1.0 days). For between-person models, <5 days per week served as reference. Wear time of “5 days per week” was defined as 5.0–5.99, “6 days per week” as 6.0–6.99, and “Worn Every Day” as 7 days of wear each week, with one-day exception in first week to account for activation technicalities. * *p* < 0.05, ** *p* < 0.01, and *** *p* < 0.001.

**Figure 4 sensors-25-02437-f004:**
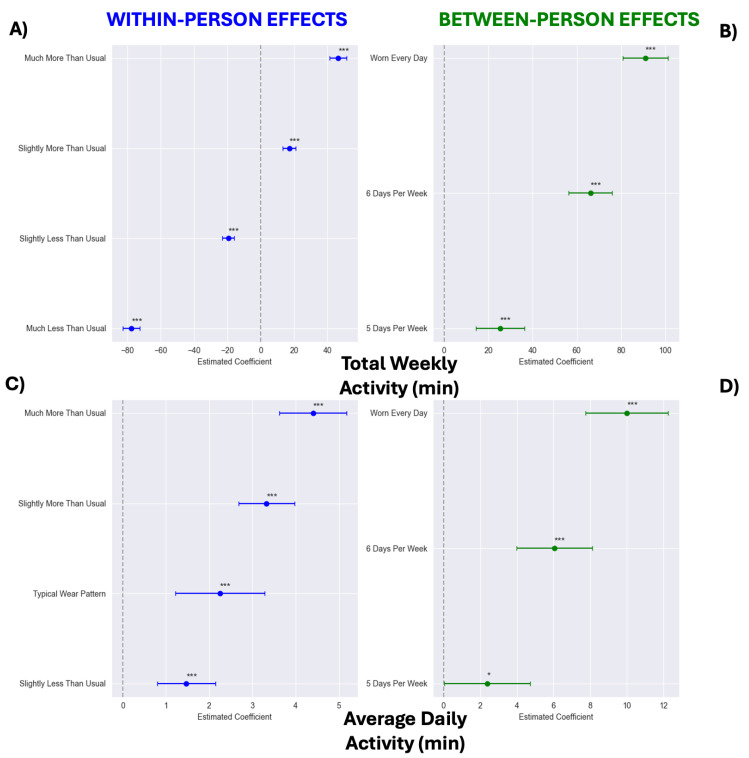
Between- (panels (**B**,**D**)) and within-person (panels (**A**,**C**)) effect estimates from categorical models of wearing WHOOP on total weekly activity minutes (panels (**A**,**B**)) and daily activity minutes (panels (**C**,**D**)). Within-person reference categories modeled as follows: “Typical Wear”, wear frequency within ±0.1 of typical (−0.1 ≤ deviation ≤ 0.1); “Much Less Than Usual”, wear frequency 2 days fewer than typical (≤−2.0 days); “Slightly Less Than Usual”, wear frequency between 2.0 and 0.1 days fewer than typical (−2.0 < deviation < −0.1 days); “Slightly More Than Usual”, wear frequency between 0.101 and 1.0 days more than typical (0.101 ≤ deviation ≤ 1.0 days); “Much More Than Usual”, wear frequency 1.0 days more than typical (>1.0 days). For between-person models, <5 days per week served as reference. Wear time of “5 days per week” was defined as 5.0–5.99, “6 days per week” as 6.0–6.99, and “Worn Every Day” as 7 days of wear each week, with one-day exception in first week to account for activation technicalities. * *p* < 0.05 and *** *p* < 0.001.

**Figure 5 sensors-25-02437-f005:**
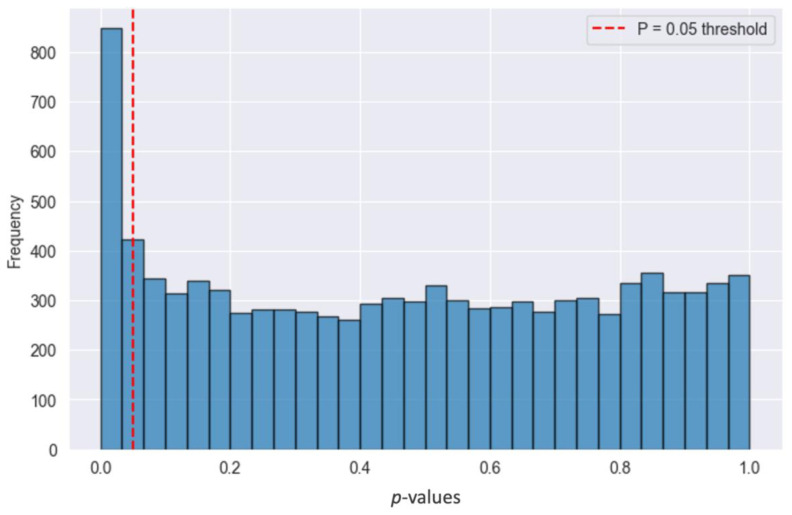
A frequency histogram showing the distribution of *p*-values from the Granger causality analysis examining whether the 7-day rolling average of WHOOP wear time predicts the next-day resting heart rate in participants who wore WHOOP for an average of 5 days per week or less (n = 1993). The analysis was conducted with a maximum lag of five days, with all lag periods included in the visualization.

**Table 1 sensors-25-02437-t001:** Participant characteristics grouped by average weekly wear time.

	<5 Days/Week	5 Days/Week	6 Days/Week	Worn Every Day
**Descriptives**				
Criteria (days/week)	<5	5.0–5.99	6.0–6.99	7.0
Weekday percentage (%)	73.77 ± 22.84 *	72.74 ± 14.58 *	71.74 ± 6.56 *	71.30 ± 1.71 *
Number of members (n)	677	1316	5570	4351
Percent male (%)	45.9	47.4	50.9 *	50.2 *
Age (yrs)	31.83 ± 11.02	31.59 ± 10.82	32.76 ± 10.97 ^	33.47 ± 11.06 *
BMI (kg/m^2^)	25.62 ± 4.91	25.67 ± 5.10	25.49 ± 4.84	25.27 ± 4.59 ^
**Baseline Biometrics**				
Resting heart rate (bpm)	64.09 ± 9.48	63.44 ± 9.19	61.80 ± 9.00 *	60.47 ± 8.63 *
Heart rate variability (ms)	56.52 ± 27.73	56.36 ± 28.44	57.02 ± 28.70	58.08 ± 29.58
**Baseline Sleep Characteristics**				
Sleep duration (hrs)	6.18 ± 1.38 *	6.44 ± 1.26 *	6.58 ± 1.13 *	6.79 ± 1.04 *
Sleep consistency (%)	57.74 ± 15.66 *	60.69 ± 15.68 *	64.34 ± 14.18 *	69.10 ± 11.9 *
**Baseline Physical Activity Variables**				
Total weekly activity (min)	151.1 ± 197.5 *	175.3 ± 197.9 *	207.4 ± 210.6 *	237.5 ± 213.5 *
Daily activity (min)	28.21 ± 36.00 *	30.42 ± 34.25 *	34.49 ± 34.95 *	38.37 ± 34.55 *

* *p* < 0.05 vs. all; ^ *p* < 0.05 vs. 5 days/week. In edge cases where members were missing their first day of data (an artifact of activation timing) they were still classified into the “Worn Every Day” category. Baseline biometric, sleep, and activity variables from the first week of the study period were measured, with the exception of sleep consistency, which was characterized in the second week of use owing to an initial calibration period.

**Table 2 sensors-25-02437-t002:** Continuous model results for biometrics.

Predictor	β	95% CI	*p*-Value
**RHR**			
Intercept	9.467	[8.787, 10.147]	<0.001
Sex [T.Male]	−0.620	[−0.724, −0.515]	<0.001
Time (Weeks)	0.144	[0.071, 0.216]	<0.001
Average Days Worn (Between-Person)	−0.441	[−0.515, −0.368]	<0.001
Time × Average Days Worn	−0.018	[−0.029, −0.007]	0.001
Person-Mean Days Worn (Within-Person)	−0.369	[−0.391, −0.347]	<0.001
Baseline RHR	0.896	[0.890, 0.902]	<0.001
Age	0.006	[0.002, 0.011]	0.005
BMI	0.030	[0.019, 0.041]	<0.001
Season [T.Spring]	0.074	[−0.013, 0.161]	0.094
Season [T.Summer]	−0.139	[−0.218, −0.061]	<0.001
Season [T.Winter]	0.226	[0.149, 0.302]	<0.001
Weekday Percentage	−0.014	[−0.016, −0.012]	<0.001
**HRV**			
Intercept	3.251	[1.737, 4.765]	<0.001
Sex [T.Male]	0.345	[0.089, 0.601]	0.008
Time (Weeks)	−0.032	[−0.213, 0.148]	0.727
Average Days Worn (Between-Person)	0.289	[0.108, 0.471]	0.002
Time × Average Days Worn	0.002	[−0.026, 0.029]	0.902
Person-Mean Days Worn (Within-Person)	0.252	[0.201, 0.303]	<0.001
Baseline HRV	0.934	[0.929, 0.939]	<0.001
Age	−0.085	[−0.097, −0.072]	<0.001
BMI	0.011	[−0.016, 0.038]	0.409
Season [T.Spring]	−0.215	[−0.425, −0.004]	0.046
Season [T.Summer]	0.469	[0.278, 0.661]	<0.001
Season [T.Winter]	−0.436	[−0.621, −0.251]	<0.001
Weekday Percentage	0.018	[0.012, 0.023]	<0.001

**Table 3 sensors-25-02437-t003:** Continuous model results for sleep variables.

Predictor	β	95% CI	*p*-Value
**Sleep Duration**			
Intercept	2.425	[2.299, 2.551]	<0.0001
Sex [T.Male]	−0.134	[−0.153, −0.116]	<0.001
Time (Weeks)	0.000	[−0.013, 0.013]	0.956
Average Days Worn (Between-Person)	0.111	[0.097, 0.126]	<0.001
Time × Average Days Worn	−0.000	[−0.002, 0.002]	0.920
Person-Mean Days Worn (Within-Person)	0.050	[0.045, 0.055]	<0.001
Baseline Sleep Duration	0.590	[0.581, 0.598]	<0.001
Age	−0.004	[−0.004, −0.003]	<0.001
BMI	−0.009	[−0.011, −0.007]	<0.001
Season [T.Spring]	−0.005	[−0.022, 0.012]	0.577
Season [T.Summer]	−0.010	[−0.025, 0.006]	0.230
Season [T.Winter]	0.033	[0.017, 0.048]	<0.001
Weekday Percentage	−0.000	[−0.001, 0.000]	0.759
**Sleep Consistency**			
Intercept	2.835	[1.168, 4.502]	0.001
Sex [T.Male]	−0.676	[−0.902, −0.449]	<0.001
Time (Weeks)	0.288	[0.080, 0.497]	0.007
Average Days Worn (Between-Person)	2.036	[1.823, 2.250]	<0.001
Time × Average Days Worn	−0.051	[−0.083, −0.019]	0.002
Person-Mean Days Worn (Within-Person)	1.144	[1.074, 1.214]	<0.001
Baseline Sleep Consistency	0.674	[0.665, 0.682]	<0.001
Age	0.074	[0.064, 0.084]	<0.001
BMI	−0.084	[−0.108, −0.060]	<0.001
Season [T.Spring]	0.087	[−0.137, 0.311]	0.446
Season [T.Summer]	0.007	[−0.194, 0.207]	0.949
Season [T.Winter]	−0.006	[−0.207, 0.195]	0.955
Weekday Percentage	0.066	[0.059, 0.074]	<0.001

**Table 4 sensors-25-02437-t004:** Continuous model results for activity variables.

Predictor	β	95% CI	*p*-Value
**Total Weekly Activity Minutes**			
Intercept	−80.035	[−102.882, 57.188]	<0.001
Sex [T.Male]	6.938	[3.093, 10.784]	<0.001
Time (Weeks)	3.471	[0.852, 6.089]	0.009
Average Days Worn (Between-Person)	33.944	[31.009, 36.880]	<0.001
Time × Average Days Worn	−1.121	[−1.522, −0.719]	<0.001
Person-Mean Days Worn (Within-Person)	26.183	[25.386, 26.979]	<0.001
Baseline Weekly Activity Minutes	0.695	[0.685, 0.705]	<0.001
Age	−0.055	[−0.228, 0.119]	0.539
BMI	−2.422	[−2.829, −2.016]	<0.001
Season [T.Spring]	−3.832	[−7.143, −0.520]	0.023
Season [T.Summer]	0.991	[−2.022, 4.003]	0.519
Season [T.Winter]	−34.438	[−37.341, −31.535]	<0.001
Weekday Percentage	−0.057	[−0.141, 0.027]	0.185
**Daily Activity Minutes**			
Intercept	2.626	[−1.944, 7.195]	0.260
Sex [Male]	1.310	[0.633, 1.986]	<0.001
Time (Weeks)	0.632	[0.186, 1.077]	0.005
Average Days Worn (Between-Person)	3.525	[2.909, 4.140]	<0.001
Time × Average Days Worn	−0.169	[−0.238, −0.101]	<0.001
Person-Mean Days Worn (Within-Person)	1.007	[0.876, 1.137]	<0.001
Baseline Daily Activity Minutes	0.527	[0.517, 0.537]	<0.001
Age	0.000	[−0.031, 0.031]	1.000
BMI	−0.467	[−0.538, −0.396]	<0.001
Season [T.Spring]	0.183	[−0.385, 0.750]	0.528
Season [T.Summer]	1.149	[0.638, 1.660]	<0.001
Season [T.Winter]	−5.088	[−5.573, −4.602]	<0.001
Weekday Percentage	−0.006	[−0.019, 0.008]	0.401

## Data Availability

The data that support the findings of this study are not publicly available due to intellectual property concerns of WHOOP, Inc. The data may be made available upon request to the company via research@whoop.com for researchers who meet the criteria for access to confidential data.

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
