# Peer review of "Wearing WHOOP More Frequently Is Associated with Better Biometrics and Healthier Sleep and Activity Patterns"

_sensors, 2025, doi:10.3390/s25082437_

Round 1
Reviewer 1 Report
Comments and Suggestions for Authors
First of all, did all 11,914 subscribers agree their data to be used in the research?
Technically, two suggestions:
1. try to analyze the principle/reason of the result/conclusion
2. detail the techniques of how to get the parameters, such as RHR, HRV, sleep etc, to convince reader that the parameters are accurate, so the conclusion in this paper is solid.
Author Response
Please see attachment, thank you.

Reviewer 2 Report
Comments and Suggestions for Authors The authors presented interesting paper on using wrist-worn wearable device for health monitoring. The mass implementation of gadgets capable of health monitoring makes it relevant to develop methods and approaches to their use for health maintenance. In the presented article, the authors showed the positive effect of frequent use of such gadgets on some aspects of health. This is a really interesting result, as opinions on this matter are mixed. The methods and design of the study are well thought out and the article is well written.I have some comments:
1) The HRV estimated from the photosensor is more properly called pulse rate variability (PRV). A great deal of work has been devoted to the issue of comparing PRV and HRV and this aspect is debatable. In ideal conditions indeed many authors demonstrate quite high similarity (but not identity) of estimates obtained from HRV and PRV. However, when vascular biomechanical properties change (in pathologies or due to external factors), it causes a significant increase in the variability of the delay between beat-to-beat events in HRV and PRV, chaotically altering estimates, including spectral and other.
2) Specify what algorithms were used to calculate sleep metrics. Was there verification?
3) Why was only one measure of HRV used?
4) At the end of the paper, discuss in detail the possible limitations of the study.
Author Response
Please see the attachment, thank you.

Round 2
Reviewer 1 Report
Comments and Suggestions for Authors
I am willing to accept the paper
Reviewer 2 Report
Comments and Suggestions for Authors
I approve revised version of paper.